# ADAM-Mediated Signalling Pathways in Gastrointestinal Cancer Formation

**DOI:** 10.3390/ijms21145133

**Published:** 2020-07-20

**Authors:** Neele Schumacher, Stefan Rose-John, Dirk Schmidt-Arras

**Affiliations:** Institute of Biochemistry, Christian-Albrechts-University, 24118 Kiel, Germany; nschumacher@biochem.uni-kiel.de (N.S.); rosejohn@biochem.uni-kiel.de (S.R.-J.)

**Keywords:** ADAM, protease, EGFR, tumour micro-environment, Notch, IL-6

## Abstract

Tumour growth is not solely driven by tumour cell-intrinsic mechanisms, but also depends on paracrine signals provided by the tumour micro-environment. These signals comprise cytokines and growth factors that are synthesized as trans-membrane proteins and need to be liberated by limited proteolysis also termed ectodomain shedding. Members of the family of A disintegrin and metalloproteases (ADAM) are major mediators of ectodomain shedding and therefore initiators of paracrine signal transduction. In this review, we summarize the current knowledge on how ADAM proteases on tumour cells but also on cells of the tumour micro-environment contribute to the formation of gastrointestinal tumours, and discuss how these processes can be exploited pharmacologically.

## 1. Introduction

Gastrointestinal organs are composed of highly complex tissue and most of these tissues are constantly replenished by the proliferation and differentiation of multi-potent tissue stem cells. These processes are regulated by paracrine signals provided by cells of the stem cell niche. Moreover, signals emitted from inflammatory cells recruited to damaged or infected tissue contribute to the mobilisation of tissue stem cells, regeneration and remodelling of the damaged tissue. Several paracrine signal proteins are synthesized as membrane-bound proteins and need to be proteolytically processed to give rise to soluble factors that then can act in a paracrine or even an endocrine fashion. Additionally, receptor molecules and proteins of the extracellular matrix are subject to proteolytic remodelling. Members of the A disintegrin and metalloprotease (ADAM) family are involved in all these processes. Aberrant activation of paracrine signal transduction is key to the development of gastrointestinal tumours and it is therefore not surprising that ADAM proteases play a decisive role in gastrointestinal tumorigenesis. Here, we give an overview of ADAM protease biology and the current knowledge regarding their tumour-promoting role in gastrointestinal malignancies. We further discuss how ADAM proteases can be therapeutically targeted for the treatment of gastrointestinal cancers.

## 2. ADAM Proteases

### 2.1. Overview

Limited ectodomain proteolysis is a regulatory mechanism mediated by proteases that converts membrane-bound proteins irreversibly into their soluble isoforms. The family of A disintegrin and metalloproteases (ADAMs) consists of 21 members (Table 1) with 13 of them being proteolytically active [1]. ADAM proteases belong to the superfamily of zinc-proteases that is characterised by the presence of an invariant HEXXHXXGXXH zinc-binding motif with the catalytic domain [2,3]. ADAM proteases are expressed as inactive zymogens, consisting of an N-terminal signal sequence, a pro-domain, a catalytic metalloproteinase domain, a disintegrin domain and a cysteine-rich, EGF-like domain followed by a transmembrane part and cytoplasmic tail (Figure 1a). In place of the EGF-like, cysteine-rich domain, ADAM17 and its close relative ADAM10 are characterized by a membrane-proximal domain in their extracellular region, which has also been reported to be involved in substrate recognition [4,5]. The prodomain has chaperone-like functions, inhibits enzymatic activity and is removed during maturation by the intracellular proprotein-convertase furin (Figure 1b) with additional proprotein convertase cleavage sites recently identified [6,7,8].

### 2.2. Regulation of ADAM Protease Activity

Proteolytic activity of ADAM proteases can either be constitutive or activated by signal transduction pathways. Regulation of ADAM protease activity is best described for the family members ADAM10 and ADAM17. Members of the membrane multi-pass families inactive rhomboids (iRhoms) [9,10] and tetraspanins (Tspans) [11,12] are involved in maturation of ADAMs throughout the secretory pathway (Figure 1b).

The rhomboid family comprises several transmembrane proteins that are evolutionarily conserved. The inactive rhomboid members lack the amino-acid residues that are indispensable for catalytic activity.

ADAM17 needs to be located at the plasma membrane in order to exert its proteolytic activity. An indispensable factor for ADAM17 activity is the inactive rhomboid protease (iRhom) 2, encoded by the *rhomboid family member* (*RHBDF*) *2* gene. iRhom2 is localized in the ER and shuttles ADAM17 from the ER to the Golgi apparatus [9]. Importantly, it is specific for ADAM17, since the transport of ADAM10 to the cell membrane is not affected by the absence of iRhom2 [10]. Whereas iRhom2 is proposed to be a myeloid-specific regulator of ADAM17 activation, its closest homologue iRhom1 is more broadly expressed [13]. The cleavage of ADAM17 substrates is seriously impaired in embryonic fibroblasts derived from iRhom2-deficient mice. However, not all ADAM17 substrates are affected by an iRhom2 deficiency, indicating a role of iRhom2 in ADAM17 substrate selection and discrimination [14].

As the name indicates, tetraspanins span the membrane four times, resulting in the formation of a short and a large extracellular loop. Tspans have the capacity to form large interacting networks, the so-called tetraspanin-enrichted microdomains (TEMs) which regulate signal transduction pathways similar to lipid rafts [15,16]. ADAM10 was previously identified to interact with members of the TspanC8 family within the ER which promoted maturation of ADAM10 to the plasma membrane [11,12]. Therefore, Tspans might emerge in general as regulators and facilitators of ADAM protease activity.

The intracellular region of ADAM17 contains numerous phosphorylation sites with phosphorylation of threonine 735 leading to rapid translocation of ADAM17 to the cell surface, resulting in increased ADAM17 activity [17,18]. It is described that the mitogen-activated protein (MAP) kinases ERK and p38 are involved in ADAM17 phosphorylation thus regulating its activity [19,20], as well as family members of the protein kinase C (PKC) family (Figure 1b). Activation of PKCs can result from engagement of G-protein coupled receptors (GPCRs) that are coupled to an αq/11 subunit resulting in phospholipase C (PLC)-mediated increase in second messengers diacylglycerol (DAG), inositol triphosphate (IP_3_) and Ca^2+^ [21,22].

Proteolytic activity of ADAM10 and ADAM17 is also triggered by exposure to negatively charged phosphatidylserine [23,24]. Phosphatidylserine in the outer leaflet of the cell membrane usually indicates cells which undergo apoptosis and cell death was indeed identified as a potent activator of ADAM proteases [25]. Moreover, ADAM protease activity can be regulated by limited proteolysis and ADAM proteases were suggested to be cleaved by other ADAM family members and the astacin family member meprin β [26,27,28,29].

The regulation of other ADAM family members is less well understood. The key to the activation of all ADAM family members is the removal of the pro-domain by proprotein convertases, as described above. However, the pro-domain of the family member ADAM8 is removed in an autocatalytic process [30]. Expression of ADAM8 itself is regulated by inflammatory cytokines such as TNFα, IL-1β, IL-4 and IL-13 [30]. ADAM12 is stored as active protease intracellularly and substrate exposure occurs via trafficking to the plasma membrane [31].

### 2.3. Signalling Pathways Regulated by ADAM Proteases

ADAM proteases play a decisive role in inflammation and cancer due to their variety of substrates. Among these substrates are cytokines, growth factors and cell surface receptors (Figure 1c). The most prominent substrate for ADAM17 is Tumor Necrosis Factor α (TNFα), a cytokine which binds to TNFα receptors (TNFRs) 1 and 2 which both are ADAM17 substrates as well [32]. The resulting soluble TNFR ectodomains can still bind TNFα thus act as antagonistic decoy receptors which has been used therapeutically. In contrast to the antagonistic activity of soluble TNFRs in TNFα signalling, the proteolytically cleaved receptor for the pleiotropic cytokine Interleukin-6 (IL-6) acts in an agonistic fashion. IL-6 signals via a membrane-bound alpha-receptor (IL-6R) and a homodimer of the signal-transducing beta-receptor subunit glycoprotein 130 (gp130), which is ubiquitously expressed, while expression of IL-6R is limited to selected cell types, e.g., hepatocytes and leukocytes. This process is termed IL-6 classic signalling. However, cells that do not express the membrane-bound IL-6R are still responsive to IL-6 via IL-6 trans-signalling. In this process the complex of IL-6 and soluble IL-6R (sIL-6R) which can be generated via limited proteolysis mediated by ADAM10 and ADAM17, is able to engage gp130 homodimers on target cells [33,34,35].

ADAM protease activity was also demonstrated to increase bioavailability of epidermal growth factor receptor (EGFR) ligands like transforming growth factor α (TGFα), heparin-bound EGF (HB-EGF) and amphiregulin (AR) [36,37,38]. EGFR signalling is crucial for regenerative processes as it was, e.g., demonstrated that both EGFR-hypomorphic mice (Waved-2) and ADAM17-hypomorphic mice (ADAM17^ex/ex^) display impaired intestinal epithelial regeneration [39,40]. The enhanced release of EGFR ligands via ADAM17 was shown to drive cancer progression and drug resistance in cell lines and experimental mouse models [41,42,43] (see also below).

Signal transduction via the Notch receptor is an evolutionary conserved pathway in which the Notch receptor undergoes a series of proteolytic events. Notch signalling was demonstrated to be essentially involved in stem cell maintenance and differentiation, but also in tumour formation [44,45]. In the majority of tissues analysed, ADAM10 was identified as the major extracellular protease for Notch [46,47]. This cleavage is followed by regulated intramembrane proteolysis (RIP) by the γ-secretase complex in order to release an intracellular domain (Figure 1c) that translocates to the nucleus and induces expression of Notch target genes [44,45]. Next to ADAM10, ADAM17 is discussed to promote Notch signalling [48,49].

Additionally, ADAM17 is described to be responsible for the proteolysis of Notch1 receptor, which increases EGFR expression [49]. The Notch receptor undergoes a series of proteolytic events in order to release an intracellular domain which is necessary for signal transduction. However, in this process the involvement of both proteases, ADAM10 and ADAM17, is discussed and seems to be context dependent [48].

## 3. ADAM-Mediated Pathways in Gastrointestinal Tumour Formation

### 3.1. ADAM Proteases in Gastric Cancer

Gastric cancer is the fourth most frequently occurring but the second most deadly cancer worldwide. One major risk for the development of gastric cancer is infection by *Helicobacter pylori* which is reflected by the fact that gastric cancer incidence is increased in areas with high *H. pylori* prevalence. Transactivation of EGFR family members and chronic inflammation are potential mechanisms how *H. pylori* infection contribute to the development of gastric cancer [50]. The vast majority of gastric cancers are adenocarcinoma arising from glandular epithelial cells of the gastric mucosa. Gastric atrophy, i.e., the loss of glandular tissue, including parietal cells, can progress to metaplasia due to the absence of signals regulating stem cell proliferation and differentiation provided by parietal cells [51]. Chronic inflammation of the gastric mucosa, e.g., triggered by bacterial infections such as *H.pylori* often results in the formation of pre-malignant lesions. Accordingly, mice with hyperactive gp130, the signal transducing subunit of the interleukin 6 (IL-6) receptor complex resulted in an excessive inflammatory response of the gastric epithelium and the formation of gastric metaplasia and spontaneous adenoma [52].

The most common genetic alterations in gastric cancer comprise among others, mutations in genes encoding for the tumour suppressor p53, the small GTPase KRAS, phosphoinositide-3 kinase (PI3K), but also members of the epidermal growth factor receptor (EGFR) family, including EGFR and ErbB2 [53,54]. KRAS, PI3K and EGFR proteins are connected within a common signal transduction pathway, highlight the importance of the EGFR/MAPK pathway for the development of gastric cancer.

Several ADAM proteases including *ADAM9*, *10*, *12*, *15*, *17* and *33* were shown to be overexpressed in gastric cancer [55,56,57,58,59,60], while *ADAM23* was shown to be epigenetically silenced by promoter CpG methylation in gastric tumour tissue [61]. Expression of *ADAM10* and *ADAM17* was demonstrated to be induced by *H. pylori* infection [55,62] and expression of *ADAM17* correlated with the forkhead transcription factor FoxM1 which is activated by PI3K/AKT [60]. ADAM17 activity is not solely dependent on its transcription level but needs additional stimuli. It is therefore not surprising that the enhanced release of EGFR ligands amphiregulin and HB-EGF upon *H. pylori* infection was dependent on the phosphorylation of ADAM17 C-terminus [62].

As described above, ligands for the EGFR can be liberated by limited proteolysis mainly via ADAM10 or ADAM17 activity. It is therefore plausible that ADAM proteases provide soluble ligands for the activation of EGFR family members on gastric tumour cells. It was indeed demonstrated that the inflammatory cytokines IL-1β and IL-8 [63], but TGFβ [64] also induced EGFR trans-activation in gastric cancer cells, particularly via soluble amphiregulin and HB-EGF (Figure 2a). Depletion of ADAM10 but not ADAM12 or ADAM17 abrogated the release of soluble EGFR ligands and hence EGFR trans-activation [65]. Upon proteolytic release of HB-EGF ectodomain, a C-terminal fragment (CTF) remains in the signal-sending cell and it was suggested that nuclear translocation of HB-EGF CTF contributes to the pathogenicity of gastric cancer [63].

Mouse models with activating mutations in *KRAS* in gastric epithelial cells but not *Helicobacter* infections resulted in an increase in AR and HB-EGF secretion, supporting the existence of a RTK/MAPK feed-forward loop in gastric cancer. However, while aberrant KRAS activation alone induced gastric metaplasia [66], the formation of adenoma was dependent on the additional activation of gp130 [54], mainly through interleukin 11 (IL-11) [67]. While in vitro experiments showed that IL-6 trans-signalling can overcome the repressive effects of trefoil factor 1 (TFF1) on IL-6 classic-signalling [68], gastric metaplasia induced by a hyperactive gp130 variant was not abrogated by a dimerised form of soluble gp130 (sgp130Fc) [69]. These data suggest that in gastric cancer ADAM10 and ADAM17 rather have a major role in EGFR trans-activation but not in boosting signal transduction via IL-6 family cytokines as in other gastrointestinal tumours (see below). 

ADAM17 was also shown to be overexpressed in gastrointestinal stroma tumours (GIST) where it co-localised with EGF and EGFR [70]. Gastrointestinal stroma tumours are rare malignant tumours of non-epithelial origin and the most common mesenchymal neoplasms of the gastrointestinal tract [71]. The majority of GISTs localise to the stomach and they are thought to originate from interstitial cells of Cajal [71], a cell type that under physiological conditions controls gastric smooth muscle cell contraction. GISTs are characterised by the occurrence of activating mutations in genes coding for the platelet derived growth factor receptor alpha (*PDGFRA*) or the close homologue c-Kit [Schmidt-Arras and Böhmer Trends Mol Med, *in press*]. PDGFRβ was previously shown to induce ADAM17 activation [72] and it is likely that ADAM17 activity in cells is elevated by mutant PDGFRα or c-Kit, resulting in the additional activation of EGFR. However, experimental evidence is still lacking.

The progression of gastric cancer correlates with the expression of *ADAM10* and *ADAM17* and high *ADAM10*/*ADAM17* expression levels were associated with patient’s poor prognosis [58,59] and the establishment of lymph node metastasis [73]. A recent report demonstrated that inflammatory cytokines IL-1α, IL-1β or TNFα secreted by diffuse-type gastric cancer cells induced upregulation of rhomboid 5 homolog 2 (RHBDF2), also termed iRhom2 in cancer-associated fibroblasts (CAFs). iRhom2 promoted ADAM17 activity and TGFβ receptor cleavage (Figure 2a) and as a consequence CAF motility to invade extracellular matrix and lymphatic vessels [74]. Hence, progression of gastric cancer is not only promoted by ADAM17 on tumour cells, but also on cells of the tumour micro-environment.

### 3.2. ADAM Proteases in Colorectal Cancer

The intestine is a complex organ comprising multiple types of differentiated and specialized cell types. The intestinal lumen is lined by a single layer of columnar epithelial cels which are constantly replenished by proliferative crypt cells. In the adult, Lgr5^+^ columnar base crypt (CBC) stem cells locate to the crypt base and give rise to the multiple intestinal cell types. The stem cell niche is formed and maintained by Paneth cells that intersperse the CBCs and provide Wnt ligands, the Notch ligands delta like (Dll) 1 and 4 and EGF. Upon exiting the stem cell niche, CBCs give rise to rapidly proliferating transit-amplifying (TA) progenitors and, subsequently, to either absorptive progenitors or secretory progenitors, dependent on Notch receptor activity. High Notch activity results in absorptive cells while low Notch activity induces secretory differentiation [75]. As outlined above, ADAM10 is the major sheddase for the Notch receptor in multiples tissues and genetic deficiency of ADAM10 in villin-expressing intestinal cells demonstrated its importance for intestinal stem cell fitness [47]. In contrast, while complete loss of ADAM10 in Paneth cells did not alter crypt homeostasis [76], overexpression of a catalytic inactive ADAM10 variant resulted in Paneth cell mislocalisation, partially phenocopying *EphB3*^-/-^ mice [77].

Proliferation of CBCs and TA progenitor cells is driven by EGFR activation [78]. Surprisingly, albeit ADAM17 plays a major role for the release of EGFR ligands, loss of ADAM17 did not influence intestinal crypt homeostasis. However, under conditions of epithelial regeneration, loss of ADAM17 in intestinal epithelial cells but not in myeloid cells impaired intestinal regeneration which was associated with impaired EGFR activation [40,79].

Intestinal tumorigenesis follows a typical adenoma–carcinoma sequence. The persistent activation of the Wnt signalling pathway is a major hallmark in intestinal cancer formation. Wnt ligands bind to its receptor Frizzled which results in stabilisation and activation of β-catenin which translocates to the nucleus to induce gene expression. In the unliganded state, β-catenin is bound in a protein complex also containing the protein adenomatous polyposis coli (APC) which is essential to induce phosphorylation of β-catenin by glycogen synthase kinase 3 (GSK3) β and subsequent proteasomal degradation of β-catenin. Loss of APC occurs often, particularly in colon cancer, and results in ligand-independent β-catenin activation. The loss of APC alleles in mice is commonly used as a model for intestinal adenoma formation. Other frequently found and early arising mutations in colorectal cancer are loss of the tumour suppressor p53 and aberrant KRAS activation [80].

Genetic deficiency of ADAM10 in intestinal stem cells drastically reduced colonic and small intestinal adenoma formation in mice with biallelic loss of APC which was linked to the absence of Notch signalling [75], indicating that ADAM10 is a potential attractive target for intestinal cancer therapy (Figure 2b). 

ADAM17 has a dual role during intestinal cancer formation. On one hand, ADAM17 releases EGFR ligands on intestinal epithelial cells and contributes to autocrine EGFR trans-activation [81]. In addition, there is evidence that the release of EGFR ligands from cancer associated macrophages depends on ADAM17 and it was demonstrated that myeloid EGFR activation (Figure 2b) is a prerequisite for the release of tumour-promoting IL-6 [82]. On the other hand, ADAM17 provides the soluble IL-6 receptor (sIL-6R) from myeloid cells that in consequence induces IL-6 trans-signalling on intestinal epithelial cells (Figure 2b) that was demonstrated to be essential for intestinal tumorigenesis [43,83]. Furthermore, there is evidence that ADAM17 on intestinal tumour cells promotes tumour angiogenesis through enhanced vascular endothelial growth factor (VEGF)-A secretion [84].

Although there is strong experimental evidence that ADAM proteases, in particular ADAM10 and ADAM17 are involved in intestinal tumorigenesis, further work is warranted to translate these findings to the human situation.

### 3.3. ADAM Proteases in Pancreatic Cancer

The pancreas is an organ with exo- and endocrine functions which synthesizes digestive enzymes such as tryptic proteases and hormones like insulin and glucagon. The most frequently occurring type of pancreatic cancer is the pancreatic adenocarcinoma within the exocrine tissue. Almost all of these cancers arise from the pancreatic ductal epithelium and are therefore termed pancreatic ductal adenocarcinoma (PDAC), while the second most common type arises from acinar cells. The head of the pancreas, which lies close to the duodenum, is the most common localisation of pancreatic cancer, and tumours in this region can result in obstructions of the pancreatic or biliary tract which is the reason why patients at diagnosis present signs of jaundice.

Pancreatic cancer mainly occurs at patients aged over 40 years and common risk factors are smoking, obesity, chronic pancreatitis, which is often linked to excessive alcohol consumption, and genetic predisposition.

PDAC is thought to arise from pre-cancerous lesions, that, depending on their localisation, can be termed pancreatic intraepithelial neoplasia (PanIN), intraductal papillary mucinous neoplasia (IPMN), pancreatic mucinous cystic neoplasia (MCN) or intraductal tubulopapillary neoplasia. Although the pre-cancerous lesions differ in their occurrence of mutations, most common mutations in pancreatic cancer are activating mutations in *KRAS*, *TP53*-deficiency and mutations in *SMAD4* [85].

Experimental models have demonstrated the importance of EGFR activation in *KRAS*-mutant pancreatic cancer [86,87], indicating that RAS/MAPK pathway activation is not sufficient to drive pancreatic tumorigenesis. This also suggests a potential critical role of ADAM17 on tumour or stroma cells to provide EGFR ligands (Figure 2c). Progression of pancreatic cancer is indeed associated with increased expression of *ADAM17* [88], and *ADAM17* expression on PDAC cells was shown to be induced by deoxycholic acid, resulting in enhanced release of the EGFR ligands AR and transforming growth factor (TGF) α [89]. Consequently, PDAC formation in a *KRAS* and *TP53*-driven mouse model of pancreatic cancer was reduced by the use of an ADAM17-directed antibody [90].

Furthermore, ADAM17 might be additionally involved in the generation of inflammatory signalling, notably IL-6 trans-signalling through the generation of sIL-6R (Figure 2c). IL-6 trans-signalling was demonstrated to promote pancreatitis-associated lung injury [91] and the progression of pancreatic intraepithelial neoplasms [92]. However, direct experimental evidence for ADAM17 in an early inflammatory stage of pancreatic cancer is still lacking.

Poor prognosis of PDAC also correlates with high expression levels of *ADAM8* and *ADAM9* [93,94]. While enhanced expression of *ADAM9* in PDAC cell lines facilitated anchorage-independent growth and was associated with increased vascularisation in a xenograft model, growth of tumour tissue was unaltered [94]. In contrast, a peptidomimetic inhibitor of ADAM8 reduced tumour growth of pancreatic tumour cells in a xenograft model but was also able to impair tumour growth in a *KRAS*-driven pancreas cancer model and significantly prolonged overall survival [95]. The pro-tumorigenic effect of ADAM8 was linked to its association with β1-integrin and the resulting increase in tumour cell motility, invasiveness and activation of the MAP kinases ERK1/2 [95]. Hence, ADAM8 represents a novel target for the treatment of pancreatic cancer [30].

### 3.4. ADAM Proteases in Hepatic Cancer

The liver is an organ with major functions in metabolism, innate immunity and detoxification. It serves as a first line defence against intestinal pathogens. It is therefore not surprising that the liver has a tremendous potential to regenerate. The liver consists of different resident cell types, including hepatocytes, cholangiocytes, Kupffer cells, hepatic stellate cells (HSCs) and liver sinusoidal endothelial cell. Hepatocytes, the hepatic epithelial cells make up the majority of cells in the liver. Upon acute hepatic damage, hepatocytes have the capacity to restore the lost liver tissue via proliferation and hypertrophy [96]. However, during chronic damage, hepatocytes undergo senescence and are unable to proliferate. Under these conditions, liver stem and progenitor cells (LPCs) proliferate and differentiate either into hepatocytes or cholangiocytes, the biliary epithelial cells [97]. ADAM10 was demonstrated to be essential for hepatocellular homeostasis via regulation of bile acid transporters [98]. Interestingly, in the liver ADAM10 is dispensable for Notch processing and ADAM10-deficient mice display normal formation of the biliary tree [98] which was linked to Notch2 activity [99].

Hepatocellular carcinoma (HCC) is one of the most frequently occurring tumour entities worldwide. Fibrotic livers can progress to liver cirrhosis which predisposes to HCC formation. The differentiation of HSCs into collagen-secreting myofibroblasts is key to fibrosis and liver cirrhosis development [100,101]. There are several indications that ADAM-induced signalling pathways are involved in both the induction and the prevention of liver fibrosis. This has been recently reviewed in depth elsewhere [102,103]. The release of EGFR ligands by HSCs can have pro- as well as anti-fibrotic activities, depending on the ligand, while the release of TNFα promotes liver fibrosis [103]. Consistent with a pro-fibrotic role of ADAM17, substrates of ADAM17 are elevated in the serum of patients suffering from liver cirrhosis [104]. In contrast, ADAM17 seems also to prevent the exacerbation of pro-fibrotic signalling (Figure 2d). HSCs lacking *Rhbdf2* that is required for ADAM17 maturation, displayed reduced TNFR1 and 2 shedding and as a consequence of enhanced TNF signalling, *Rhbdf2^-/-^* mice displayed enhanced bile duct obstruction-induced liver fibrosis [104]. Increased expression of *ADAM12* was detected in activated HSCs and in liver tissue sections from patients suffering from liver cirrhosis or HCC [105,106], suggesting that ADAM12 has a tumour-promoting role in the liver by actively remodelling the extracellular matrix.

*ADAM10* was shown to be overexpressed in HCC and associated with a poor prognosis [107]. miRNAs targeting ADAM10 reduced the invasive potential of HCC cell lines [108,109]. While direct experimental evidence is still lacking, these findings suggest that ADAM10 has a tumour promoting role in the liver, potentially via induction of Notch signalling (Figure 2d), which was suggested to promote cholangiocarcinoma [110].

EGFR signalling has a dual role in the liver. EGFR signalling on hepatocytes was demonstrated to enhance intrinsic DNA damage repair and chronic inflammatory signalling in obese mice via IL-1β and TNFα abrogates this tumour suppressing effect of EGFR (Figure 2d) [111]. In contrast, similar to its role in intestinal tumour formation, autocrine activation of EGFR on Kupffer cells was demonstrated to promote IL-6 secretion (Figure 2d) [112]. The development of HCC in humans was associated with elevated serum levels of IL-6 [113,114], and murine models have demonstrated that HCC formation is blunted in the absence of IL-6 [115]. Interestingly, HCC formation seems to be completely dependent on IL-6 trans-signalling (Figure 2d) and beside IL-6, Kupffer cells also provide sIL-6R [116]. Although experimental evidence is still lacking, it is tempting to speculate that ADAM17 on Kupffer cells is essential to (i) release EGFR ligands to induce IL-6 release and (ii) generate sIL-6R in order to promote tumorigenesis via IL-6 trans-signalling.

## 4. ADAM-Directed Therapeutic Approaches

Due to its major role in TNF processing, ADAM17 has previously gained interest as therapeutic target for the treatment of chronic inflammatory diseases and has led to the development of a series of small molecular inhibitors. The majority of these inhibitors is hydroxamate-based (Figure 3a) and relatively specific for ADAM17 or dual inhibitors for ADAM10 and ADAM17. These inhibitors bind to the zinc ion in the catalytic centre and to the S1’ pocket of ADAM17 [117,118,119]. However, none of the earlier inhibitors reached beyond phase 3 in clinical trial due to musculoskeletal and liver toxicities [118]. This led to the development of more specific inhibitors such as INCB7839, a dual inhibitor of ADAM10 and ADAM17 which has been used in clinical trials for the treatment of HER2^+^ breast cancer, large B-cell non-Hodgkin lymphoma [118] and is currently in clinical testing for the treatment of pediatric glioma. These data will also show if severe side effects encountered by use of the earlier inhibitors is linked to their unspecificity or to the fact that ADAM10/17 have a broad substrate spectrum with proteins involved in different biological processes.

Another approach to target ADAM activity is the development of antibodies targeting the catalytic site and the use of recombinant pro-domains of ADAM proteases (Figure 3b,c). Two monoclonal antibodies D1 (A12) [90,120,121] and MEDI3622 [122] directed against ADAM17 were demonstrated to be efficient in pre-clinical cancer models. The monoclonal antibody 8C7 was directed against an active conformation of ADAM10 and was shown to inhibit lymphoma growth in a xenograft model and gastrointestinal tumour growth in a genetic mouse model [123].

The pro-domains of ADAM proteases share only a low level of similarity and are thought be highly specific for their cognate protease. The use of recombinant pro-domains has therefore been considered as an alternative approach to specifically target the catalytic site of ADAM proteases and the inhibitory selectivity was demonstrated for pro-domains of ADAM10, 12 and 17 [7,117,124,125]. ADAM17 recombinant pro-domain has been shown to specifically inhibit ADAM17 in different mouse models [8,126,127,128] and might enter into clinical development for the treatment of inflammatory bowel disease [118].

Peptidomimetics were shown to be useful to inhibit ADAM8 multimerisation and hence autoactivation in pancreatic cancer [95]. This approach might also be applicable to other ADAM family members, where the association with integrins enhances cancer cell motility.

Overexpression of ADAM family members on tumour cells make them an attractive target for immunotherapies. An ADAM17/CD3 bispecific T-cell enganger (BiTE) antibody was able to induce the T-cell mediated lysis of pancreatic cancer cells in vitro (Figure 3d) [129]. Furthermore, peptides derived from ADAM17 were found to be present on major histocompatibility complex (MHC) I molecules on prostate cancer cells and therefore represent an immunotherapeutic target [130]. However, ADAM protease-directed immunotherapy is only beginning to evolve and definitely warrants further investigation.

## 5. Conclusions

There is now a growing body of evidence that members of the ADAM protease family are over-expressed on tumour cells and are correlated with tumour formation and the progression of gastrointestinal tumours (summarized in Table 2). Some of the tumorigenic signalling pathways initiated by ADAM proteases have been deciphered. However, unbiased proteomics approaches are needed to identify ADAM substrates and ADAM protease-activated signalling pathways that might represent novel targets for tumour therapy.

Given the fact that ADAM proteases generate paracrine signals, the role of ADAM proteases in the tumour microenvironment needs more attention. Cell type-specific gene targeting in mouse models of cancer combined with unbiased proteomic substrate identification will improve our understanding of how ADAM proteases shape the tumour microenvironment and promote tumour metastasis. This will also help to design tissue-specific ADAM protease inhibitors or to target ADAM-mediated novel tumour cell vulnerabilities, with the aim to reduce the severe side effects encountered with current ADAM protease inhibitors.

## Figures and Tables

**Figure 1 ijms-21-05133-f001:**
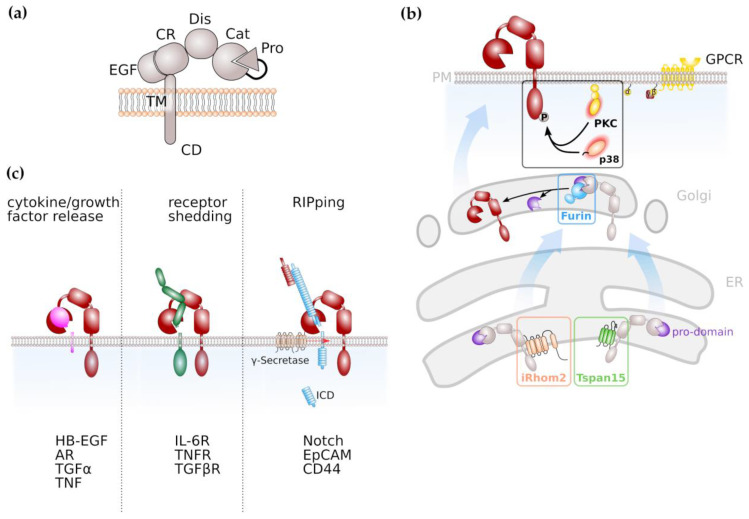
ADAMs and ADAM-mediated signalling pathways. (**a**) General domain structure of ADAM proteases. Pro, pro-domain; Cat;catalytic domain; Dis, disintegrin domain; CR, cysteine-rich domain; EGF, EGF-like domain; TM, transmembrane domain; CD, cytoplasmic domain (**b**) Maturation of ADAM protease, notably ADAM10 and ADAM17 is facilitated by tetraspanins (Tspans) and inactive rhomboid proteases (iRhom), respectively. Activation of ADAM proteases, notably ADAM17, via C-terminal phosphorylation is mediated by protein kinase C (PKC) or the mitogen-activated kinase p38. (**c**) ADAM proteases have diverse substrates and are involved in cytokine/growth factor release, shedding of receptor molecules and the induction of intracellular signal transduction via limited proteolysis, followed by regulated intramembrane proteolysis (RIP).

**Figure 2 ijms-21-05133-f002:**
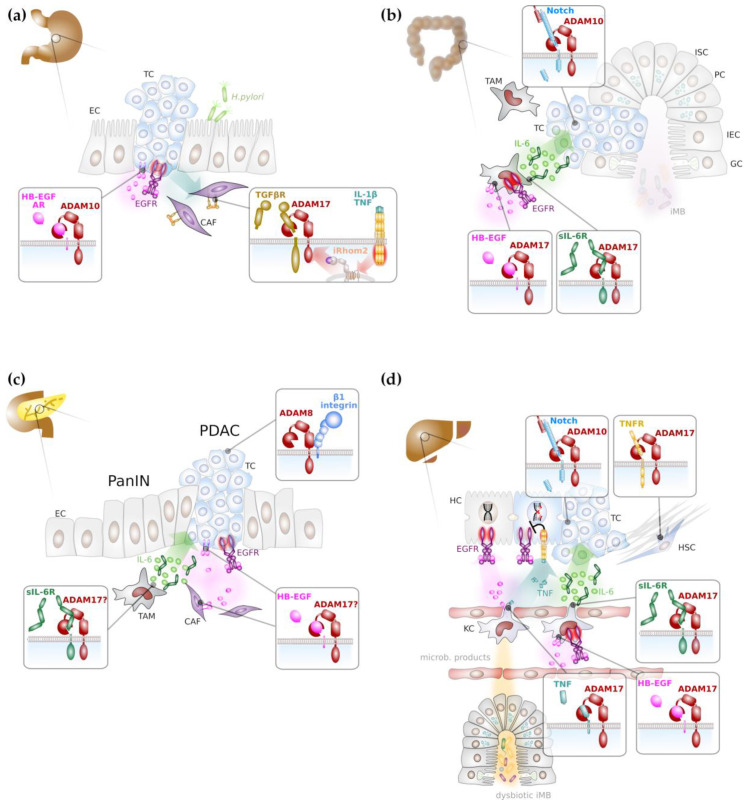
Major ADAM-mediated signalling pathways in gastrointestinal cancer formation. (**a**) ADAM proteases in particular ADAM10 and ADAM17 promote gastric tumorigenesis mainly through activation of epidermal growth factor (EGF) receptor (EGFR) signalling on cancer cells and through inhibition of transforming growth factor (TGF) β receptor (TGFβR), thereby enhancing motility of cancer-associated fibroblasts (in purple). (**b**) Colonic tumorigenesis depends on Notch signalling promoted by ADAM10. In addition, myeloid ADAM17 is supposed to promote inflammatory signalling via proteolytic release of EGFR ligands, such as heparin-bound EGF (HB-EGF) and concomitant induction of interleukin (IL) 6 (green circles). In concert with the soluble IL-6 receptor (sIL-6R), IL-6 induces trans-signalling on colon epithelial cells. (**c**) EGFR signalling and IL-6 trans-signalling promotes progression of pancreatic intraepithelial neoplasia (PanIN) to pancreatic ductal adenocarcinoma (PDAC), presumably via ADAM17 activity. (**d**) DNA damage response (DDR) in hepatocytes is enhanced via EGFR signalling. Intestinal dysbiosis induces tumour necrosis factor (TNF) secretion from Kupffer cells (KCs), the liver-resident macrophages that impairs the beneficial effects of EGFR signalling on DDR. Both, EGFR ligands and TNF are presumably released by ADAM17 from KCs. Liver fibrosis is reduced by ADAM17 via impaired TNF receptor signalling in hepatic stellate cells. Hepatocarcinogenesis is promoted by IL-6 trans-signalling that is presumably induced by ADAM17 on KCs. Notch signalling mediated by ADAM10 in hepatocytes promotes hepatic tumorigenesis. Abbreviations: CAF: cancer associated fibroblast; EC: epithelial cell; GC: goblet cell; HSC: hepatic stellate cell; IEC: intestinal epithelial cell; iMB: intestinal microbiome ISC: intestinal stem cell; KC: Kupffer cell; PC: Paneth cell; TAM: tumour associated macrophage; TC: tumour cell.

**Figure 3 ijms-21-05133-f003:**
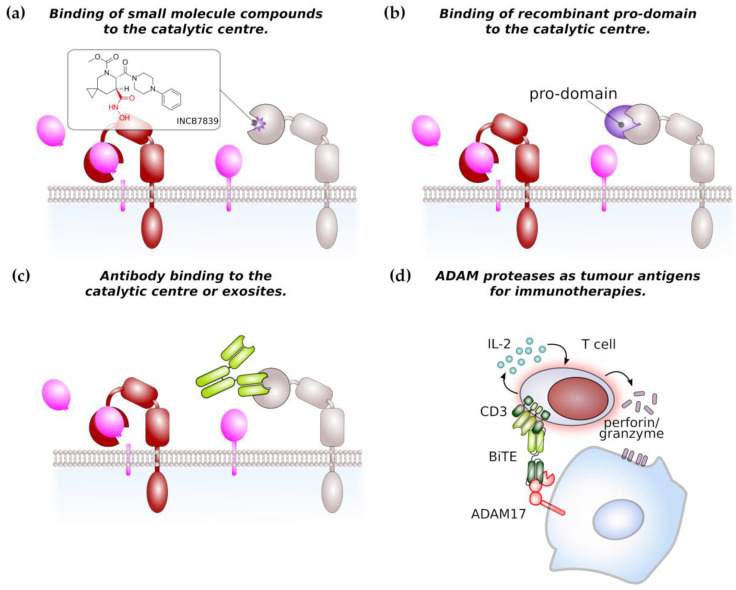
ADAM proteases as potential therapeutic target in cancer therapy. (**a**) Most of small molecule compounds targeting ADAM protease catalytic domain are based on a hydroxamate structure (in red). The structure of the ADM17-specific inhibitor INCB7839 is shown as an example. (**b**) Recombinant pro-domain binds to the catalytic centre of its cognate ADAM protease thereby limiting access to the catalytic domain. Inhibition by pro-domains is considered to be highly specific. (**c**) Monoclonal antibodies targeting the catalytic site or regulatory exosites of ADAM proteases impair their catalytic activity. (**d**) ADAM protease can be target of immunotherapies. As an example, a bispecific T-cell engager (BiTE) single-chain antibody against CD3 and ADAM17 induces the T-cell-mediated killing of pancreatic cancer cells in vitro. Arrows indicate secretion of IL-2 or perforin/granzyme.

**Table 1 ijms-21-05133-t001:** Classification of the human ADAM protease family.

**Proteolytically Active (HEXXHXXGXXH)**	testis specific		ADAM 20,21,30
	not testis specific	haematopoietic	ADAM 8,28
		non haematopoietic	ADAM 9,10,12,15,17,19,33,DEC1
**Proteolytically Inactive**	testis specific		ADAM 2,7, 18,29
	not testis specific		ADAM 11,22,23,32

**Table 2 ijms-21-05133-t002:** Overview of ADAM protease activity in gastroenterological tumours.

	ADAM ProTeases	Signals UpStream of ADAMs	Pathways Activated by ADAMs		Biological Effect
**Gastric Cancer**	ADAM9, 10,12,15,17,33**↑**	IL-1β, IL-8, TGFβ	TC	AR, HB-EGF/EGFR**↑**	tumour growth
	ADAM17	PDGFR/c-Kit			
	ADAM17	IL-1β, TNF/iRhom2	CAF	TGFβR**↓**	CAF, TC motility
**Colorectal Cancer**	ADAM10	n.i.	TC	Notch signalling**↑**	block in differentiation, adenoma formation
	ADAM17	n.i.	TC	IL-6/gp130 trans-signalling**↑**	tumour growth
		n.i.	EC	VEGF/VEGFR**↑**	vascularisation
**Pancreatic Cancer**	ADAM8	n.i.	TC	β1 integrin/ERK1/2	tumour growth, TC motility
	ADAM9	n.i.	TC	integrins, HB-EGF/EGFR	vascularisation
	ADAM17	deoxycholic acid	TC, SC	AR, TGFα/EGFR**↑**	PDAC formation
		n.i.	TAM/TC	IL-6/gp130 trans-signalling**↑**	PaNIN progression
**Hepatic Cancer**	ADAM10	n.i.	TC	Notch signalling	tumour progression
	ADAM17	dysbiotic iMB	KC/HC	TNF/TNFR1/2**↑**	reduced DNA damage repair
	ADAM17	n.i.	KC/TC	IL-6/gp130 trans-signalling**↑**	HCC formation
	ADAM17	iRhom2	HSC	TNF/TNFR1/2**↓**	limiting fibrosis

Abbreviations used: CAF: cancer associated fibroblast, EC: endothelial cells, HC: hepatocyte, HSC: hepatic stellate cell, KC: Kupffer cell, TC: tumour cell, n.i.: not identified, all other abbreviations are spelled out in the main text. **↑** indicates up-regulated; ↓ indicates down-regulated.

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
