# Peer review of "ADAM-Mediated Signalling Pathways in Gastrointestinal Cancer Formation"

_ijms, 2020, doi:10.3390/ijms21145133_

Round 1

Reviewer 1 Report

Schumacher et al wrote a review describing the ADAM-mediated signalling pathways in gastrointestinal cancer formation. The manuscript gives a general introduction on the regulation of ADAM family member and the signal pathways that they are involved in. Moreover, the manuscript reviews the ADMA-mediated pathways in gastrointestinal cancer formation, including gastric cancer, colorectal cancer, pancreatic cancer and hepatic cancer. The authors also discuss the ADAM-directed therapeutic options in clinical application. Overall, the review is well-organized with a clear structure and is focused on the recent updates of ADAM family member in gastrointestinal cancer formation, which could be of interest to many readers.

Here are some suggestions for the authors to consider:

  1. It would be better to give a brief overview of the ADAM family member in point 2.1, for example, name all the 21 members (ADAM1, 2, 7, 8, ….).
  2. In point 2.2 the authors mainly reviewed the regulation of ADAM10 and ADAM17, is there any literature on other ADAM family members?
  3. It would be better to add a future direction and conclusion section to summarize and discuss the future direction on ADAM family members.
  4. It would be much clearer if the authors could summarize the role of ADAM family members in different cancer formation in a table.

Author Response

We highly appreciate the reviewer’s comments and constructive suggestions to which we would like to respond point by point as follows:

Here are some suggestions for the authors to consider:

It would be better to give a brief overview of the ADAM family member in point 2.1, for example, name all the 21 members (ADAM1, 2, 7, 8, ….).

We thank the reviewer for this constructive suggestion. We have now included a table summarizing all 21 ADAM members and refer to the table within section 2.1

In point 2.2 the authors mainly reviewed the regulation of ADAM10 and ADAM17, is there any literature on other ADAM family members?

Regulation of other ADAM family members is less well understood. We have included an additional short paragraph in section 2.2 summarizing activation of ADAM8 and ADAM12.

It would be better to add a future direction and conclusion section to summarize and discuss the future direction on ADAM family members.

We have now added a short outlook as section 5 at the end of the manuscript.

It would be much clearer if the authors could summarize the role of ADAM family members in different cancer formation in a table.

This is a very good suggestions. Accordingly, we have now included a table at the end of section 3 summarizing the role of ADAM proteases in gastrointestinal cancer.

Reviewer 2 Report

Schumacher et al's manuscript titled "ADAM-mediated signalling pathways in gastrointestinal formation" discusses ADAM proteases mediated signaling pathways leading to the formation and progression of gastrointestinal cancers.

Role of ADAM proteases in cancer has been extensively reviewed in the past https://doi.org/10.1186/1559-0275-8-9. The current review focusses on the  role of ADAM proteases in gastric cancers and suggests therapeutic potential of these molecules. While the review makes an interesting read, the authors need to address the following comments:

  1. Fig 1: indicate iRhom, Tspan15, Furin in bold and bright colors.
  2. Fig 2A: legible labels and figure is needed to show inflammatory cytokines activating iRhom2 leading to the activation of ADAM17. the inlet box showing ADAM10 is slightly irrelevant here as ADAM 17 is the focus of gastric cancer discussed here.
  3. Fig 2B: label the different components legibly.
  4. Fig 2C: show the role of ADAM8 as it is mentioned as a therapeutic target in line 289.
  5. Figs 3A,B,C require labels as it is not intuitive to decipher the mode of action of these molecules as therapeutic targets
  6. line 27: extracellular matrix are subject "to" 
  7. Line 28: members of the "A" disintegrin
  8. line 115: "but"also in tumor formation
  9. line 313 talks about Rhbdf2. what is Rhbdf2? explain. 
  10. the review talks about the role of different ADAM molecules in gastric, hepatic, colorectal and pancreatic cancer. the authors do not comment on the role of ADAM inhibitors that are already in use against breast, lung cancer to name few. Combination of ADAM inhibitors with standard of care chemotherapy agents are already in use. 
  11. since ADAM 10 and ADAm 17 act on a wide variety of membrane proteins, what is the implication if they are targeted therapeutically?

Author Response

We are grateful to the reviewer for his careful reading and his constructive suggestions. We reply to the issues raised point by point as follows:

Fig 1: indicate iRhom, Tspan15, Furin in bold and bright colors.

In order to improve the labelling we have changed it to bold font and highlighted activation pathways using frames.

Fig 2A: legible labels and figure is needed to show inflammatory cytokines activating iRhom2 leading to the activation of ADAM17. the inlet box showing ADAM10 is slightly irrelevant here as ADAM 17 is the focus of gastric cancer discussed here.

We have improved labelling of Figure 2. As by the reviewer’s suggestion we have included a box demonstrating cytokine-mediated upregulation of ADAM17 via iRhom2 in Figure 2 A.

Fig 2B: label the different components legibly.

We have extended and improved the labelling of all subfigures in Figure 2.

Fig 2C: show the role of ADAM8 as it is mentioned as a therapeutic target in line 289.

We have now included ADAM8 into the Figure.

Figs 3A,B,C require labels as it is not intuitive to decipher the mode of action of these molecules as therapeutic targets

We have added headings to all of the subfigures in order to render the mode of action directly visible.

line 27: extracellular matrix are subject "to"

Line 28: members of the "A" disintegrin

line 115: "but"also in tumor formation

We are grateful to the reviewer for spotting these errors. They have been corrected accordingly.

line 313 talks about Rhbdf2. what is Rhbdf2? explain.

We are grateful to the reviewer for spotting this inconsistency. We have added an explanatory sentence at the introduction at line 69. This paragraph reads now like, with changes highlighted in red:

An indispensable factor for ADAM17 activity is the inactive rhomboid protease (iRhom) 2, encoded by the rhomboid family member (RHBDF) 2 gene. iRhom2 is localized in the ER and shuttles ADAM17 from the ER to the Golgi apparatus [9]”

the review talks about the role of different ADAM molecules in gastric, hepatic, colorectal and pancreatic cancer. the authors do not comment on the role of ADAM inhibitors that are already in use against breast, lung cancer to name few. Combination of ADAM inhibitors with standard of care chemotherapy agents are already in use.

To the best of our knowledge, hydroxamate inhibitors of ADAM17 are the best developed inhibitors. So far, none of them received clinical approval. We are not aware of ADAM inhibitors that are routinely in clinical use and also did not find any literature reporting routine use of specific ADAM inhibitors in the clinic.

since ADAM 10 and ADAm 17 act on a wide variety of membrane proteins, what is the implication if they are targeted therapeutically?

Targeting ADAM10/17 is a challenging task as most of the inhibitors display severe side effects. We already mentioned this fact in section 4. As it is not clear if this is linked to unspecificity or to the fact that these ADAMs have a spectrum of diverse substrate proteins, we have added the following sentence in section 4:

"These data will also show if severe side effects encountered by the earlier inhibitors is linked to their unspecificity or to the fact that ADAM10/17 have a broad substrate spectrum with proteins involved in different biological processes."